# Comprehensive Attention Self-Distillation for Weakly-Supervised Object Detection - A Reproducibility Study

## Reproducibility Summary

**Scope of Reproducibility**

We perform extensive ablation studies reproducing the results of the paper "Comprehensive Attention Self-Distillation for Weakly-Supervised Object Detection." In this paper the authors propose a method of regularisation via aggregating attention maps to improve the accuracy of weakly supervised object detectors. They propose to aggregate attention maps from different layers of the network, and across different views of the same image, and use these maps to encourage features with better object coverage. The paper claims that this allows them to train an object detector in a weakly supervised manner and achieves state-of-the-art results.

**Methodology**

Using the official code released by the authors we have re-run the ablation experiments on the Pascal VOC 2007 dataset using our own GPU compute resources. We made changes to the released code to further investigate the effects of other model components, extending the ablation studies. We have also performed an in-depth analysis of the code itself to assess code quality for ease of modification and reuse.

**Results**

We found that we were able to meet or exceed the published results. Further ablation studies showed that we were able to achieve the same results without the proposed attention regularisation. Our results do not agree with the original paper's claims that the novel regularisation provides an improved object detector. We performed further ablation studies to identify the source of the improvement, attributing it to the ContextLocNet-style head, Inverted Attention module, regression branch, and stronger data augmentation.

**What was easy**

The fundamental concept is simple and intuitive. The figures in the paper work to make it even clearer, and provide a strong motivation as to why this might be useful. Some parts of the codebase are reused from other public weakly-supervised object detection codebases, anyone familiar with those works will have an easier time following this one.

**What was difficult**

There was significant difficulty in working with the code due to poor coding practices, poorly documented requirements, and errors in the code. Experimentation was limited due to the slow training process, which owes both to the computational cost of this approach as well as some inefficient implementation choices.

**Communication with original authors**

We had no direct communication with the original authors. We browsed public communications via GitHub issues posted to the public code repository.

# 1  Introduction

Weakly Supervised Object Detection attempts to train object detection models utilising only image-level class labels (as opposed to the typical bounding-box level annotations). This alleviates the high annotation cost associated with constructing quality object detection datasets. Since the training annotations are less granular than the desired model predictions, this problem is fundamentally ill-posed and many strategies have been proposed to overcome it.

The authors propose that there is a degree of noise introduced into the training process due to the inconsistency between model predictions on different views of the same image. In the fully supervised setting, if you train a model with multi-scale inputs, flips, etc. the box annotations are transformed alongside the image. In the weakly supervised setting, which typically generates box-level pseudo-annotations from its own predictions as a self-training procedure, there is no guarantee that the same annotations would be selected in different scales or flips of an image. They acknowledge that typically the pseudo-annotations are incomplete, and that different views often identify different instances or object parts, all of which are desirable to identify. As such the authors propose to aggregate information between views and across network layers into a maximal attention map, which they then use to regularise the network to encourage consistent attention between views whilst also increasing object coverage.

# 2  Methodology

## 2.1  Implementation(s)

We utilised the published implementation by the authors for the majority of our experiments. We first performed an extensive examination of the implementation to understand precise details of the method that are not discussed in the paper. We found that the published code had some errors that prevent it from running as of the time of writing. We fixed these issues to the best of our ability in our implementation, noting all changes in the source.

We also made changes for the purposes of our experiments. We instrumented the codebase with the Weights and Biases platform introduced by Biewald [2020] for experiment tracking and analysis. We also added some configuration flags, and utilised them to enable and disable model components by configuration for our ablation studies.

## 2.2  Dataset

We selected Pascal VOC 2007 introduced by Everingham et al. [2007] as the dataset on which to perform our ablation studies. VOC07 is a popular choice for ablation studies amongst weakly supervised works as it is relatively small, and so more experiments can be completed on a limited time/resource budget.

As is standard practice we omit annotations labelled "difficult", and during evaluation we ignore detection of these instances. We trained the model on the union of the training and validation sets ('trainval' containing 5011 images) and evaluate on the public test set. This setting is the same as the ablation studies performed in the original paper, and is the standard format used in prior weakly-supervised object detection works.

## 2.3  Hyperparameters

We use the same hyperparameters as the original paper for all experiments.

## 2.4  Resources

We utilised the two NVIDIA Titan RTX GPUs recommended by the original authors for all experiments.

## 2.5  Model Breakdown

To perform our ablation studies we need to define the individual components of the network and training procedure.

### 2.5.1  Initial Configurations

The original paper identifies eight configurations in their ablation studies. They provide a baseline, three progressively more complex versions of Image-wise CASD, Layer-wise CASD, combined Image-wise and Layer-wise (i.e. 'Full') CASD, as well as adding a regression branch and extra augmentations to the full version.

### 2.5.2 Additional Configurations

By analysing the code we identify two other components that are not evaluated in the ablation studies. We first note that the model makes use of the pseudo-labelling algorithm introduced by Tang et al. [2018a] for all experiments. To assess general applicability we also consider the simpler maximum scoring pseudo-labelling algorithm presented in Tang et al. [2017].

We also note that rather than the multiple instance head introduced by Bilen and Vedaldi [2016] that is typical of OICR-like weakly supervised object detection models, the authors utilise the ContextLocNet Contrastive-S model from Kantorov et al. [2016]. This head is significantly more computationally expensive, but produces better initial seed labels than WSDDN.

For the additional configurations we only consider their impact on the baseline, and full versions of CASD.

## 3 Code Analysis

An in-depth review of the code was performed to both identify any key details that are absent from the paper, as well as to determine the ease of reuse and modification of the work for others that wish to reproduce the results or build on top of them. It is important that research code is of high quality to ensure correctness of the results as well as reproducible science.

### 3.1 Initial Setup

The initial setup required only some small tweaks to ensure the program launched. Some required libraries are included with version information, such as Python, PyTorch, and CUDA. Others are listed as required but absent version numbers mean that the most current version in the repositories may not be suitable. At time of writing this was the case with the 'scipy' library, which is currently at version 1.7 but requires a version below 1.3. Another library is entirely absent from the requirements, 'easydict'. Finally the listed version of CUDA is in-fact incorrect, causing unexpected and somewhat cryptic errors with library loading. As of time of writing the listed CUDA requirement is CUDA 10.0, however the actual requirement is 9.0.

Although not especially difficult to debug, it would be preferable to include tested version numbers for as many libraries and packages as possible. At a minimum the Python package versions should be stated for all required packages, if not managed by a virtual environment package list such as pipenv or conda. It is also essential that when version numbers are listed they are correct.

### 3.2 Implementation Bugs

There are several issues which prevent the program from running as expected or running at all. There are multiple cases where arguments are passed incorrectly, either with the wrong types or in the wrong places, resulting in crashes. Anyone desiring to leverage this work will have to fix these issues before being able to run the code.

There are also user arguments passed in when running an experiment that are overwritten by program code. The provided shell script for training requires the user select a GPU by providing an index, however in the python script that is subsequently launched to perform training, the selection is overwritten with the constant value "0,1" (corresponding to the first two GPUs on the system). This may cause issues in multi-tenancy systems or if other experiments are running, since the code forces the usage of GPUs regardless of the selection of the user.

Another issue with the public implementation is that Image-wise CASD is missing from the loss calculation in the latest version of the public code. During the loss calculation step Image-wise CASD is calculated, however it is never added to the total loss value. As such the implementation does not actually make use of it without a user modifying the code to add it back into the final loss summation, along with it's corresponding weighting factor $\gamma$.

### 3.3 Missing Components

The final major issue with the public code is that it does not appear to contain a functioning Layer-wise CASD implementation. There is present a function `ca_lw()` however it is never called in the program and as such does not contribute. Additionally the `ca_lw()` function does not appear to implement the Layer-wise CASD as described in the paper, rather it performs CASD over the image and its flip more akin to an incomplete version of Image-wise CASD.

We made best efforts to re-implement Layer-wise CASD, however due to the complexity of the codebase and the limited description in the paper we were unable to do so. As such all our experiments will lack the Layer-wise CASD component.

### 3.4 General Code Quality

A review of the public implementation revealed several issues that make it extremely difficult to understand and modify. There are several instances of no-operation or unclear lines such as `for i in range(1)` (which assigns `i` to the value `0` and executes the 'loop' once only). This adds a large cognitive overhead to working with the codebase as any potential users need to interpret what these lines are actually doing and whether they have any impact on the program flow.

The code also has a high amount of repetitiveness. There are many cases where lines or blocks of code are repeated with minor changes such as which variables are being used, index values, etc. This can make it difficult for a reader to determine whether these blocks of code are doing the same thing, or how they differ, as there may only be one or two characters difference between the lines, or they may even be the same with another variable having changed between repeated blocks. This means that if a user wishes to modify the program (for example if they want to build on top of it) they must make sure that they find all of these repeated blocks and make the same changes. Good development practice would say that these repeated blocks should be either functions for those that are the same but operating on different variables, or loops for those where the code is the same but an index variable has changed between repeated blocks.

The code also frequently switches between utilising PyTorch and NumPy for representing tensors. Both libraries provide highly similar interfaces for working with tensors, however only PyTorch includes the automatic differentiation components for performing backpropagation. Utilising both PyTorch and NumPy heavily can make it difficult to determine which type is being used to represent a given tensor at various points in the program. As mentioned in Section 3.2 the public implementation includes an error introduced by this issue, which causes the program to crash when one method expecting a NumPy array is passed a PyTorch tensor instead.

The switching between PyTorch and NumPy also lowers the computational efficiency, as only PyTorch operates on high-speed GPU devices. As such, conversions between the two involve a copy from GPU memory to system memory, and then execute on the CPU, before potentially being copied again to GPU memory. These copy operations add a time and space overhead that could be avoided in many cases by writing pure PyTorch code.

Overall the code quality issues make this work a poor choice for reuse. The code quality issues make it too easy to accidentally make changes that do not do what they are intended to, or fail to make the changes uniformly across the program.

## 4 Results

### 4.1 Reproducibility

| Method | Original | Reproduction |
|---|---|---|
| Baseline | 48.9 | 52.47 |
| Image-wise w/o IA | 52.6 | 52.47 |
| Image-wise | 54.1 | 54.12 |
| Layer-wise | 52.3 | N/A |
| Full CASD | 56.8 | 57.77 |

Table 1: Original vs Reproduced results

As can be seen in Table 1, the results of our reproduction mostly met or exceeded the original published results. The results of the baseline and Image-wise with no Inverted Attention are identical, which was unexpected. Deeper inspection of the code showed that this was because there is a flag in the code that is always set to false, disabling Image-wise CASD. Regardless of this, we were able to reproduce the Image-wise results with and without Inverted Attention to within $\pm 0.15$. Furthermore, our experiments exceeded the published result of Full CASD even though both Layer-wise and Image-wise CASD are absent. These results suggest that CASD is not essential to achieving this results, and may be a hindrance (although further experiments would need to be performed to establish this).

Due to time constraints and the slow turnaround time for experiments we were unable to re-evaluate the results after the Image-wise bug was found. We were able to run some preliminary experiments which showed that the CASD Image-wise loss was on the order of $10^{-5}$ whilst the other losses were on the order of $10^{-1}$. This suggests that the

159 impact would be extremely minimal, and this is also supported by an issue on the public GitHub that was opened by
160 another user[1].

## 4.2 General Applicability

| Pseudo-Labelling method | Method | mean Average Precision |
|---|---|---|
| OICR | Baseline | 52.00 |
| | CASD | 56.73 |
| PCL | Baseline | 52.47 |
| | CASD | 57.77 |

Table 2: CASD with different labelling functions

162 Table 2 shows the results of using the simpler top-1 pseudo-labelling algorithm that was originally introduced with
163 Tang et al. [2017]. As can be seen there is a significant improvement regardless of the pseudo-labelling method,
164 however there is slightly more when using PCL. This suggests that the changes are generic and provide improvement
165 regardless of the pseudo-labelling method used. The use of a regression branch has been covered in prior works by
166 Ren et al. [2020], Wang et al. [2018] and is well known to improve performance even under the weakly supervised
167 setting. Stronger data augmentation is also a common practice for improving image processing models of all types,
168 and is extremely well studied. The results suggest that the Inverted Attention proposed by Huang et al. [2020a] is also
169 useful for weakly-supervised object detection.

## 4.3 Effect of ContextLocNet

| Method | With | Without |
|---|---|---|
| Baseline | 52.47 | 39.38 |
| CASD | 57.77 | 45.58 |

Table 3: CASD with and without ContextLocNet-style neck and head

171 As shown in Table 3 the results drop drastically with the exclusion of the ContextLocNet-style head. The difference in
172 performance between the baseline and 'CASD' is maintained, further strengthening the argument that the improvements
173 are agnostic to the model structure. The approximately 12 point reduction does however suggest that the overall
174 accuracy is primarily driven by the inclusion of this component instead of the simpler WSDDN-style head that is used
175 in most prior works such as Ren et al. [2020], Tang et al. [2017, 2018a,b], Wang et al. [2018].

# 5 Discussion

177 By re-running the original ablation studies on our own hardware we are able to say with a high degree of confidence that
178 the overall accuracy of the work is reproducible. By utilising the author's code it is possible to train a state-of-the-art
179 weakly supervised object detection model. However, our further studies pose questions as to whether the source of the
180 improvement is as presented.

181 The results from Section 4.1 suggest that only the Inverted Attention and flip-and-scale+colour augmentation are
182 important components in achieving strong WSOD results. This is further supported by the decision of the original
183 authors to not enable the Image-wise or Layer-wise CASD regularisation in their public implementation. Unfortunately
184 due to the state of the code discussed in 3 we missed the flag disabling Image-wise CASD, and due to time constraints
185 were unable to re-run all experiments and evaluation after fixing this bug. Anecdotal evidence from our own and others'
186 experiments as discussed supports the impact being minimal, however further experiments would need to be conducted
187 to confirm this.

188 Since the public implementation lacked CASD Layer-wise completely, and we were unable to implement it ourselves,
189 we can only speculate on its efficacy. Because we were able to exceed the published final result we can show that it
190 isn't essential to achieving good results, however it is possible that it would provide further improvements. It is likely
191 though that similar to Image-wise CASD the loss value is extremely low. Both use the same method for computing the
192 regularisation term, just applied to different sets of attention maps.

---

[1] https://github.com/DeLightCMU/CASD/issues/6

The results of Section 4.3 show that the methods proposed in Kantorov et al. [2016] are a major contributing factor to the results. Although Kantorov et al. [2016] showed this method to be effective, only a small number of later works such as Wan et al. [2019b,a] utilise it. This is most likely due to the high computational and memory cost, which includes three separate passes through the fully connected layers (the most expensive parts of the network). Our results suggest that it would be valuable to reconsider the inclusion of this method in modern works, as well as considering alternatives that achieve similar results with a lower computational cost.

As discussed in Section 3 one of the largest barriers to utilising this work is the state of the public code. Improved dependency management via an environment manager such as `conda` or `pipenv` would resolve most of the dependency issues, being able to provide correct versions of all python libraries. Although the CUDA version issue may be resolvable by `conda`, when dealing with system libraries it may be easier to manage those externally, or provide a reproducible container via Docker.

Implementation bugs and missing components are a significant barrier to reproducible work. All authors should endeavour to set up and run their public implementation in a clean environment to ensure that such bugs do not exist at release. It would be ideal if these issues were caught in tests written to ensure the correctness of the released program, however this is not yet standard practice and is a problem in the wider machine learning research community.

The general code quality issues are likely a symptom of the modern 'publish or perish' mindset, where researchers of all levels are encouraged or required to publish as frequently as possible. This disincentives spending time on maintenance tasks such as refactoring code, writing tests, etc. to ensure that it is easily utilised and reproduced. Poorly written code has significant risk associated with it, as it makes mistakes more difficult to identify. We believe this is the case with CASD Image-wise being disabled in the public implementation, as it was difficult to find all the required locations to enable it.

### 5.1 What was easy

The fundamental idea of the paper is simple and intuitive. The authors explanation is clear and straightforward, with diagrams to improve clarity.

### 5.2 What was difficult

The running of experiments was extremely slow on our Titan RTX GPU(s). Although the code allows for the usage of more than one GPU, it does not do this in an efficient way and as a result receives almost no benefit from this method of training. The Inverted Attention step and the ContextLocNet-style head are also both computationally expensive operations to perform, this is unavoidable but contributes to this problem.

Experiments on the small Pascal VOC 2007 dataset took approximately 70 hours (approx. 3 days) to complete a full training cycle plus a single test. To extrapolate from this, experiments on the larger VOC 2012 dataset would take around 6 days and MS COCO would take over a week. In comparison, the public implementation of OICR can be trained to achieve approximately 53 mAP in only 8 hours on a single GPU[2].

The code quality issues discussed in 3 also made the implementation difficult to work with. Configurations that are ignored, 'magic numbers', unused functions and variables, and no-op lines make it a slow process to verify that any changes have been made in all the appropriate locations, and are complete to take effect. This challenge presented itself in this report with the absence of Image-wise CASD, since a flag is set to `false` inside the code at all times.

### 5.3 Communication with original authors

We did not have any direct communications with the authors. Several people (including us) have attempted to communicate with the authors via the GitHub Issues on the public repository, and those issues were read and considered when writing this reproducibility report.

## 6 Conclusions

This report sought to assess the reproducibility of the paper "Comprehensive Attention Self-Distillation for Weakly-Supervised Object Detection" by Huang et al. [2020b]. We found that whilst the final results could be reproduced or exceeded with the publicly released implementation, this implementation included neither a working implementation of Image-wise or Layer-wise CASD as proposed in the paper. We performed extensive ablation studies to determine

---

[2]https://github.com/ppengtang/pcl.pytorch

which components contribute most to the model performance and found that the data augmentation, regression branch, and utilisation of a constrastive weakly supervised head were all key factors. We also performed an analysis of the code, identifying many issues that make it difficult to apply to new problems, or build on top of for future works. Overall we believe that further investigation needs to be done to verify the usefulness of the Image-wise and Layer-wise CASD regularisation techniques.

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
