# OpenReview forum: "Comprehensive Attention Self-Distillation forWeakly-Supervised Object Detection - A Reproducibility Study"
_ML_Reproducibility_Challenge/2021/Fall — RC2021_

### Official Review · Reviewer_BVde · 2022-03-31
**Successful reproduction with additional ablations and insights beyond the original work**

**Rating:** 8
**Confidence:** 4

**Review:**


Reproducibility summary: Included and good

Scope of reproducibility: Included, good

Code: Combined original authors' code (which was hard to use) with other publicly available codebases

Communication with original authors: The report authors commented on the original authors' GitHub repository. but would have been very helpful, especially with the poor quality of the code. I also believe the authors of the original paper would find the ablation study extremely interesting.

Hyperparameter Search: None

Ablation Study: Yes, the paper performs an extensive suite of ablation studies that actually pinpoints the source of the improvements of the object detector. With these ablations, the paper exposes additional insight into the nature of the contributions, stating that they are due to factors beyond those just mentioned in the paper. The authors were able to achieve the same results in the original paper, even without the proposed attention regularization.

Discussion on results: The authors are confident that the results in the original paper can be reproduced after running the original ablation studies on their own hardware. While the numerical results are not exactly the same as the original paper, they are similar, and the trends are the same. Therefore, the paper successfully is able to reproduce the original work.

Recommendations for reproducibility: There is a very detailed analysis of the authors' code (Section 3). In addition, there are additional ablation studies performed to better validate the claims in the paper. With the additional ablation studies, the authors discovered insights that even the original paper authors did not include from the report (and perhaps did not realize).

Results beyond the paper: A key additional finding in this work is as to whether or not the original paper's proposed method is really what is driving the increased performance. The report considers additional ablation results (Section 4) to show that actually, they can achieve higher accuracies, even without the proposed method in the paper. This is an important contribution.

Overall organization and clarity: The report is well-organized, clear, and easy to understand.

---

### Meta-Review · Program_Chairs · 2022-04-07

**Recommendation:** Accept
**Confidence:** 4

**Metareview:**

This paper does a good job in verifying the reproducibility of the original paper, and goes above and beyond by doing an ablation study to confirm results. There are some clarity improvements to be done to make the paper more standalone, for instance by specifying hyperparameters and model breakdown so readers don't need to refer to the original paper. The analysis of the code readability was useful, and the Discussion was profound and interesting.

---

### Decision · Program_Chairs · 2022-04-09

**Decision:**

Accept

**Comment:**

Following the recommendation of reviewers and meta-reviewer, the paper is accepted for ML Reproducibility Challenge 2021, and will be published in the upcoming special edition of ReScience Journal.